# A Patient-Specific 3D+t Coronary Artery Motion Modeling Method Using Hierarchical Deformation with Electrocardiogram

**DOI:** 10.3390/s20195680

**Published:** 2020-10-05

**Authors:** Siyeop Yoon, Changhwan Yoon, Eun Ju Chun, Deukhee Lee

**Affiliations:** 1Center for Medical Robotics, Korea Institute of Science and Technology, 5, Hwarang-ro 14-gil, Seongbuk-gu, Seoul 02792, Korea; h14515@kist.re.kr; 2Division of Bio-medical Science and Technology, KIST School, Korea University of Science and Technology, Seoul 02792, Korea; 3Cardiovascular Center, Seoul National University Bundang Hospital, Seongnam 13620, Korea; kunson2@snu.ac.kr; 4Department of Radiology, Seoul National University Bundang Hospital, Seongnam 13620, Korea; humandr@snubh.org

**Keywords:** 3D+t modeling, coronary artery, non-rigid registration, cage deformation, 4D CT

## Abstract

Cardiovascular-related diseases are one of the leading causes of death worldwide. An understanding of heart movement based on images plays a vital role in assisting postoperative procedures and processes. In particular, if shape information can be provided in real-time using electrocardiogram (ECG) signal information, the corresponding heart movement information can be used for cardiovascular analysis and imaging guides during surgery. In this paper, we propose a 3D+t cardiac coronary artery model which is rendered in real-time, according to the ECG signal, where hierarchical cage-based deformation modeling is used to generate the mesh deformation used during the procedure. We match the blood vessel’s lumen obtained from the ECG-gated 3D+t CT angiography taken at multiple cardiac phases, in order to derive the optimal deformation. Splines for 3D deformation control points are used to continuously represent the obtained deformation in the multi-view, according to the ECG signal. To verify the proposed method, we compare the manually segmented lumen and the results of the proposed method for eight patients. The average distance and dice coefficient between the two models were 0.543 mm and 0.735, respectively. The required time for registration of the 3D coronary artery model was 23.53 s/model. The rendering speed to derive the model, after generating the 3D+t model, was faster than 120 FPS.

## 1. Introduction

Cardiovascular disease (CVD) is one of the primary causes of death worldwide, with 22.2 million deaths expected by 2030. According to National Health and Nutrition Examination Survey (NHANES) data from 2013 to 2016, the prevalence of CVD was 48.0% in adults over the age of 20. The prevalence of CVD has a positive correlation with an increase with age [1]. The resulting social cost is estimated to have been 351.3 billion dollars in the U.S. alone, from 2014 to 2015. In particular, cardiovascular disease accounted for 14% of total medical spending in U.S., the highest rate among other major diagnostic groups—even higher than cancer. In the global population, the burden of expenditure is even more serious [1].

Providing sufficient information through image analysis acquired in the pre-operative diagnosis stage eliminates unnecessary examination and helps in developing patient-specific treatment plans. As the heart is a continuously beating organ and there may be unexpected movements in patients (e.g., arrhythmia), if information on this movement can be obtained in advance, coronary artery and heart procedures may be more efficient.

As shown in Figure 1, the ECG signal is a change in potential that is correlated with the movement of the heart muscle. Motion of the heart produces changes in volume and pressure in the cardiac chambers; therefore, ECG provides important information about the movement of the heart. When CT is reconstructed by performing retrospective ECG synchronization, the movements of the heart and coronary arteries (according to the cardiac cycle) can be obtained geometrically, and movement information (e.g., coronary artery distortion), as well as characteristics of the coronary stenosis, can be obtained.

Patient-specific 4D heart shape information facilitates the following applications: accurate coronary artery structure acquisition [2,3], analysis of 4D blood flow and stenosis [4,5,6], removal of motion artifacts in the vascular region [7,8,9], surgical simulation for each patient [10], postoperative evaluation and analysis [5,6], atrial motion analysis [11,12], vascular motion analysis [13,14], and real-time deformation prediction during surgery combined with 2D images [8].

In particular, cardiac CT angiography (CTA) has an isotropic spatial resolution of less than 0.5 mm and, so, can be used to observe the movement of the coronary artery and trabeculae of the ventricle. In addition to grading the degree of calcification of the coronary artery and the total amount of plaque from the CT image, it is also possible to measure the torsion of the coronary artery at a high resolution [15]. This high-resolution spatial information can help the operator perform a procedure appropriate for each patient before and after surgery. However, despite the high spatial resolution of the CTA, its temporal resolution is 50–200 ms, which is lower than that of 4D echocardiography (30–100 Hz) and cardiac MRI (30–50 ms). Therefore, proper shape interpolation for restoring high time resolution information from CTA imagery is essential for co-operation with the other applications while preserving high-precision anatomical details.

One of the essential requirements of cardiac modeling in many of these applications is that the topology of the mesh model constituting the cardiac model must be consistently preserved. In particular, in the case of a 4D heart model, the mesh should not cause new problems (e.g., self-intersection or mesh degeneration), even if the positions of the vertices constituting the shape change over time. To address these requirements, many researchers have employed the template-based registration scheme.

The registration process matches a template model to a target model through geometric shape deformation. The most popular registration algorithm is the iterative closest point (ICP) method, which consists of finding the correspondence between two models and finding the optimal transformation [16]. However, the ICP method is sensitive to the initial position and noise, while the shape registration is limited up to rigid transformation. Non-rigid registration deals with the deformation of the shape in addition to rigid transformations. Non-rigid registration is more challenging, however, as non-rigid transformations not only require more correspondences to be defined, but the solution space is much more extensive [17]. Research into the registration of 3D non-rigid shapes has been actively conducted using the methods of delineating shape deformation and shape correspondence.

To describe the deformation of an object, with respect to its dynamics and material properties, many researchers have assumed shape deformation to be a physical model, such as a linear elastic model [18], non-linear elastic model [19,20], viscous fluid [21], or diffusion model [22,23]. In particular, the Large Deformation Diffeomorphic Metric Mapping (LDDMM) framework provides robust deformation as a massive flow consisting of diffeomorphisms [24]. However, the physical models are computationally expensive and sensitive to mechanical properties. On the other hand, the statistical shape deformation model (SSM) uses a low-dimensional statistical model, in which shape deformation is inferred from a training data set [25,26]. Although SSM reduces the computational cost, the shape of variability is limited by the training data. Therefore, the deformation is hardly representative of the inter-variability of patients, such as the topological discontinuity of coronary arteries. Nora et al. described the motion modeling problem using the coronary arteries attached to the SSM of muscles [27]. Due to the representation of the coronary artery, the deformation poorly described the lumen diameter. Instead of modeling a priori physical and statistical information, there have been attempts to estimate the shape transformations with landmarks and coherent motions. Radial basis function methods express shape deformations as weighted sums of distance function for control point changes [28]. In particular, the thin-plate spline method minimizes the bending energy, which has a closed-form solution [29]. The most popular form of deformation is known as B-spline free-form deformation (FFD) [26,30,31]. Rueckert et al. [32] have proved the conditions for FFD to have diffeomorphic deformation. This method has disadvantages, however: the numerical cost increases with the number of control points and the degree of freedom of shape deformation is fixed. Therefore, the deformation has limited representation capacity.

In addition to deformation modeling, establishing correspondences between shapes is a critical problem in registration. The one-to-one correspondence of the ICP method is sensitive to the initial position and shape loss. To determine many-to-many correspondence points, Chui et al. [33] used the fuzzy correspondence between two shapes. The problem of selecting a robust point matching the correspondence has been interpreted as a combination of a Gaussian mixture model (GMM) and Expectation Maximization [34]. In the GMM model, one point is the centroid of the Gaussian distribution for the points constituting the shape, while the other point is regarded as the data to generate [35]. The variations of GMM have different deformation models, according to the obtained transformation parameters and the regularization term. For example, regularizing the second derivative of the transformation leads to a thin-plate spline transformation, while regularizing according to motion coherence theory leads to a coherent point drift transformation [36,37]. The variations of GMM have been generalized using the generalized Gaussian radial basis function [38]. To represent the local spatial representation, an $L_{2}E$ estimator has been proposed, which creates a robust sparse–dense correspondence [39,40].

However, the mixture model requires a high computational cost, as it generates a Gaussian distribution for each of the points. Furthermore, each Gaussian distribution shares its standard deviation among the points. The mixture model is thus sensitive to noise and shape loss. Especially for coronary arteries, narrow and tangled structures are very challenging to model in 3D+t. This is because the loss of blood vessel morphology can be observed in different heartbeats of the same patient, through motion artifacts and geometrical deformations.

In this paper, we propose a 3D+t coronary artery model that can be inferred in real-time, according to ECG signals. The overall structure of the proposed method is shown in Figure 2. The proposed patient-specific 3D+t coronary artery motion model is divided into two processing blocks, according to the timing of data processing; (1) a preoperative processing block, and (2) an intraoperative usage block.

At the preprocessing step, we first perform the segmentation of 4D CTA volumes to generate artery models. After we have multiple coronary artery models, hierarchical cage-based registration is performed to construct a patient-specific 3D+t model with hyperelastic regularization. The proposed hierarchical cage deformation model more robustly/accurately registers coronary artery models in different cardiac phases. When updating the control points of the coronary artery model, we gradually increase the degrees of freedom of the deformation model. A modified hyper-elastic regularization term prevents mesh degeneration problems during the control point optimization step. After the optimal cage control point is obtained—which minimizes the shape dissimilarity of the source shape and the target shape—we interpolate the shape control point to build a continuous 3D+t model. The interpolated shape model provides fast shape-inference for intraoperative usage.

In the intraoperative usage step, the ECG phase can be assessed from the patient’s real-time signal, which is then correlated to the geometric deformation of cardiac muscles, as shown in Figure 1. Therefore, the 3D shape of the coronary artery is provided by a continuous 3D+t model, according to change of the ECG signal and time.

The contributions of the proposed method consist of the following:A hierarchical deformation method to perform robust shape registration, even with incomplete coronary artery models;Rapid shape interpolation that enables restoring small and complex geometry in a time-varying coronary artery model;The modified hyper-elastic regularization prevents mesh degeneration during shape registration; andEvaluation of the proposed method using retrospective data for eight patients, both qualitatively and quantitatively.

## 2. Pre-Processing ECG-Gated 4D CT Images

In this study, we reconstructed CTA volumes for eight patients at 0% to 95% intervals (in 5% intervals) between the RR peaks of the heart rate. We took the volumes using a 256-slice multi-detector CT scanner (BRILLIANCE ICT 256 SLICE, Philips Healthcare) at the Cardiovascular Center of Seoul National University Bundang Hospital. This retrospective study was approved by the Institutional Review Board of Seoul National University Bundang Hospital (IRB No. B-2009-637-103).

The cross-sectional size of the image was 512 × 512 pixels, the average number of slices in the Z-axis direction was 298, and the volume voxel resolution was 0.35 × 0.35 × 0.45 mm. The left ascending and circumflex coronary arteries were segmented using the ITK-Snap software [41]. The segmented arteries were converted to mesh models using Poisson surface reconstruction, where the average number of nodes was 10,638. We selected 75% phase mesh models as templates, as the left ascending and circumflex coronary arteries are most clearly observed at 75% phase [42]. Figure 3 shows the models of the template and other phases.

## 3. Hierarchical Cage-Based Shape Registration Method

In this section, we address the non-rigid registration method to find the optimal deformation between coronary artery models. This section is organized as follows: (1) shape representation and registration problems; (2) gradient descent for shape control point optimization; (3) multi-resolution cage deformation representation; and (4) diffeomorphism supported by hyper-elasticity regularization;

### 3.1. Shape Representation and Registration Problems

This section provides the basic concepts of representation and registration of shapes. Let a shape *V* be $V=\{v_{i}|v_{i}\in R^{3},i=0,\ldots ,n-1\}$, which contains *n* of vertices. If $V_{s}$ and $V_{t}$ are source and target shapes, respectively, the registration problem is to find an optimal transformation that minimizes the dissimilarity between the shapes. Here, an arbitrary transformation *T* maps the source shape $V_{s}$ to the target shape $V_{t}$. Through the optimization process, the optimal transformation parameters $x^{*}$ minimize disparity measure as follows:(1)$$x^{*}=\arg\min_{x}d\left(T\left(x\right)\circ V_{s},V_{t}\right).$$

The transformation *T* is a mapping such that
(2)$$\begin{array}{cc} T:V_{s}\to \bar{V} & =V_{s}+U\left(V_{s},x\right), \end{array}$$
where $\bar{V}$ is the deformed shape, *x* are the local deformation parameters, and U(V,x) is a vertex-wise mapping.

The shape transformation *T* may be represented through the modification of a coarse cage mesh that envelops the source shape. Let a region Ω bound the shape $V_{s}$ in 3D. The sub-region $\Omega_{r}$ is a sub-divisions of Ω, where $\Omega ={⋃}_{\forall r}\Omega_{r}$ and $\Omega_{i}\cap \Omega j=\emptyset$, i≠j. If we create the m×m×m regular lattice grid on the region Ω, the sub-divisions of Ω contain ${\left(m-1\right)}^{3}$ control vertices and $m^{3}$ sub-regions. Hereby, the sub-region $\Omega_{r}$ is defined as an 8-point cuboid. The eight corner points of $\Omega_{r}$ are given as $P_{r}=\left\{p_{i}|p_{i}\in R^{3},i=0,\ldots ,7\right\}$. The linear combination of cage control points and their local coordinates represent the vertices of the shape $V_{s}$. If the vertex $v\subset \Omega_{r}$, then the representation of the vertex by the sub-region control point is given as below:(3)$$v=F\left(v;P_{r}\right)=\sum_{i=0}^{7}\varphi_{i}\left(v\right)p_{i},$$
where $\phi_{i}$ is a trilinear shape function for assigning local coordinates, such as
(4)$$\begin{array}{c} \varphi_{0}\left(v_{x},v_{y},v_{z}\right)=\left(1-v_{x}\right)\left(1-v_{y}\right)\left(1-v_{z}\right)/8\\ \varphi_{1}\left(v_{x},v_{y},v_{z}\right)=\left(1+v_{x}\right)\left(1-v_{y}\right)\left(1-v_{z}\right)/8\\ \varphi_{2}\left(v_{x},v_{y},v_{z}\right)=\left(1+v_{x}\right)\left(1+v_{y}\right)\left(1-v_{z}\right)/8\\ \varphi_{3}\left(v_{x},v_{y},v_{z}\right)=\left(1-v_{x}\right)\left(1+v_{y}\right)\left(1-v_{z}\right)/8\\ \varphi_{4}\left(v_{x},v_{y},v_{z}\right)=\left(1-v_{x}\right)\left(1-v_{y}\right)\left(1+v_{z}\right)/8\\ \varphi_{5}\left(v_{x},v_{y},v_{z}\right)=\left(1+v_{x}\right)\left(1-v_{y}\right)\left(1+v_{z}\right)/8\\ \varphi_{6}\left(v_{x},v_{y},v_{z}\right)=\left(1+v_{x}\right)\left(1+v_{y}\right)\left(1+v_{z}\right)/8\\ \varphi_{7}\left(v_{x},v_{y},v_{z}\right)=\left(1-v_{x}\right)\left(1+v_{y}\right)\left(1+v_{z}\right)/8. \end{array}$$

From the previous definition of a cage representation, the motion of vertex *v* in the direction $\bar{v}$ is given as follows:$$\begin{array}{cc} u(v,P_{r}) & =\bar{v}-v\\ \; & =\sum_{i=0}^{7}\varphi_{i}\left(v\right)\left(p_{i}+\partial p_{i}\right)-\sum_{i=0}^{7}\varphi_{i}\left(v\right)p_{i}\\ \; & =\sum_{i=0}^{7}\varphi_{i}\left(v\right)\partial p_{i}, \end{array}$$
where u(v,x) is the motion of vertex *v* and $\partial p_{i}$ is the motion of control point $p_{i}$. Therefore, the shape deformation is only dependent on the change of control points, as shown in Figure 4. Therefore, the parameter of the cage representation of the transformation *T* is given as follows: (5)$$x=\{\partial p_{i}|\partial p_{i}\in R^{3},i=0,\ldots ,7\}.$$

### 3.2. Gradient Descent for Shape Control Point Optimization

The optimization of the transformation is defined as the process of minimizing a metric. If we set the disparity measure as the squared Euclidean distance between the correspondence pair, then
(6)$$\begin{array}{cc} d(v_{s},v_{s},P) & =∥T\left(P\right)\circ v_{s}-v_{t}{∥}^{2} \end{array}$$
(7)$$\begin{array}{cc} \; & =∥F\left(v_{s};P\right)-v_{t}{∥}^{2} \end{array}$$
(8)$$\begin{array}{cc} \; & =∥\sum_{i=0}^{7}\varphi_{i}\left(v_{s}\right)p_{i}-v_{t}{∥}^{2}, \end{array}$$
where $v_{s}\in V_{s}$, $v_{t}\in V_{t}$, and $v_{s},v_{t}$ is correspondence pair.

Thus, the gradient can be denoted as the sum of the difference vector of the corresponding pair multiplied by the weight of each control point. Therefore, the partial derivative of the disparity measure for a cage control point $p_{i}=\left(p_{ix},p_{iy},p_{iz}\right)$ is
(9)$$\begin{array}{cc} \frac{\partial }{\partial p_{i}}d\left(v_{s},v_{t},P\right) & =\frac{\partial }{\partial p_{i}}{∥v_{s}-v_{t}∥}^{2} \end{array}$$
(10)$$\begin{array}{cc} \; & =2∥v_{s}-v_{t}∥\cdot \frac{\partial }{\partial p_{i}}∥\sum_{i=0}^{7}\varphi_{i}\left(v_{s}\right)p_{i}-v_{t}∥ \end{array}$$
(11)$$\begin{array}{cc} \; & =2∥v_{s}-v_{t}∥\cdot \varphi_{i}\left(v_{s}\right). \end{array}$$

The Jacobian matrix for the disparity measure of one correspondence pair is
(12)$$\frac{\partial }{\partial P}d\left(v_{s},v_{t},P\right)=\left[\begin{array}{c} \frac{\partial }{\partial p_{0}}d\left(v_{s},v_{t},P\right)\\ \frac{\partial }{\partial p_{1}}d\left(v_{s},v_{t},P\right)\\ \vdots \\ \frac{\partial }{\partial p_{m}}d\left(v_{s},v_{t},P\right) \end{array}\right]=\left[\begin{array}{ccc} \partial p_{0x} & \partial p_{0y} & \partial p_{0z}\\ \partial p_{1x} & \partial p_{1y} & \partial p_{1z}\\ \; & \vdots & \\ \partial p_{mx} & \partial p_{my} & \partial p_{mz} \end{array}\right].$$

The update of cage control points uses the distribution of the differences of corresponding pairs, which are the vectors from sources to targets generated inside the sub-regions $\Omega_{r}$. At this time, the robust correspondence selection potentially supports the update of cage control points. To establish the correspondence pair robustly, we constrain the correspondence searching process using orientation filtering, as follows:(13)$$\left\{v_{s},v_{t}\right\}=\left\{\begin{array}{cc} \mathrm{Paired}, & \mathrm{if}\;\theta \left(\vec{n_{s}},\vec{n_{t}}\right)<\theta_{\mathrm{Threshold}}\\ \;\mathrm{Not}\;\mathrm{paired}, & \mathrm{otherwise} \end{array}\right.,$$
where θ(·,·) is angle between the two vectors, and $\vec{n_{s}}$ and $\vec{n_{t}}$ are the vertex normals of $v_{s}$ and $v_{t}$, respectively. We set $\theta_{\mathrm{Threshold}}=30^{\circ }$. The Jacobian matrix for the sum of the squared Euclidean distance is:(14)$$\sum_{\left\{v_{s},v_{t}\right\}\in \forall I_{V_{s}}}\frac{\partial }{\partial P}d\left(v_{s},v_{t},P\right)=\left[\begin{array}{c} \sum \partial p_{0}\\ \sum \partial p_{1}\\ \vdots \\ \sum \partial p_{m} \end{array}\right],$$
where $I_{V_{s}}$ is the set of correspondence pairs.

### 3.3. Multi-Resolution Cage Deformation Representation

In this section, we present a cage deformation method using multi-resolution to represent a gradual deformation. In the registration process, the resolution of the cage determines the degree of freedom of shape deformation. With increasing resolution of the cage, the deformation model can represent a more detailed shape change. However, a dense cage has the disadvantage that it can lose the overall shape. A method for maintaining local shape features through multi-resolution or hierarchical data structures is, thus, used as a complementary method.

We assumed the generation of the cage based on a regular lattice grid. The primitive shape of the cage obtained from the lattice structure is a cube with eight vertices and six quadrilateral faces, where the points inside the cage can be represented as linear combinations of cage control vertices. As shown in Figure 4, the cage can be partitioned into the inner sub-regions, where the control points of this sub-region can be created using the control points of the outer region. We denote the vertex *v* using cage control points $P^{n}$ at the deformation depth of *n* as follows:(15)$$v=F\left(v;P^{n}\right)=\sum_{i=0}^{7}\varphi_{i}\left(v\right)p_{i}^{n},$$
where $p_{i}^{n}$ is *i*th cage control point at the deformation depth *n*. If we recursively acquire the sub-region of the region Ω that surrounds the source model, we can denote the higher-level control points using the lower level control points. The generalized formula presents the corner points of the sub-division, which are recursively described in the multi-resolution process below:(16)$$P^{m}=F\left(P^{m};P^{n}\right)=\sum_{i=0}^{7}\varphi_{i}\left(P^{m}\right)p_{i}^{n},$$
where m>n and m,n∈N. Thus, if we represent the shape using a chain of cage deformations, the deformed vertex $\bar{v}$, with respect to the level *n* deformation, is
(17)$$\bar{v}=F\left(v;P^{n},\partial P^{n}\right)=\sum_{i=0}^{7}\varphi_{i}\left(v\right)\left(p_{i}^{n}+\partial p_{i}^{n}\right).$$

Similarly, the deformed vertex $\bar{v}$ by level *n* deformation after level n−1 deformation is
(18)$$\begin{array}{cc} \bar{v} & =F(v;P^{n},P^{n-1})\\ \; & =\sum_{i=0}^{7}\varphi_{i}\left(v\right)\left(\sum_{j=0}^{7}\varphi_{j}\left(p_{i}\right)\left(p_{j}^{n-1}+\partial p_{j}^{n-1}\right)+\partial p_{i}^{n}\right). \end{array}$$

To cooperate with the gradient descent, we reformulate $\frac{\partial }{\partial p_{i}}d\left(v_{s},v_{t},P\right)$ as a multi-resolution process. The partial derivative of the given cost function at the ${\left(n-1\right)}^{\mathrm{th}}$ level is given as follows:(19)$$\begin{array}{cc} \frac{\partial }{\partial p_{i}^{n-1}}d\left(v_{s},v_{t},P\right)\; & =\frac{\partial }{\partial p_{i}^{n-1}}{∥\sum_{i=0}^{7}\varphi_{i}\left(v_{s}\right)p_{i}^{n}-v_{t}∥}^{2}\\ \; & =\frac{\partial }{\partial p_{i}^{n-1}}{∥\sum_{i=0}^{7}\varphi_{i}\left(v_{s}\right)\sum_{j=0}^{7}\varphi_{j}\left(p_{i}^{n}\right)p_{j}^{n-1}-v_{t}∥}^{2}\\ \; & =2∥v_{s}-v_{t}∥\cdot ∥\sum_{i=0}^{7}\varphi_{i}\left(v_{s}\right)\varphi_{j}\left(p_{i}^{n}\right)∥. \end{array}$$
Modification of the control point $\partial p_{i}$ in the multi-resolution cage sub-division is carried out by
(20)$$\partial p_{i}=\partial p_{i}^{n}+\partial p_{i}^{n-1}+\cdots +\partial p_{i}^{1}.$$

### 3.4. Diffeomorphism Supported by Hyper-Elasticity Regularization

Although hierarchical cage deformation recursively represents shape deformation to avoid local minima, dense cages possibly lead to more cage degeneration. Therefore, an appropriate regularization process is required when applying hierarchical transformations. For plausible deformation, we used hyper-elastic regularization, which prevents unexpected partial deformation. We utilized and modified the study of Burger et al. [20], which can be easily extended to the cage deformation setting. As shown in Figure 5, the 24 sub-regions of the cage were defined using corner points $p_{i}$ and seven auxiliary points, which are the volume points $p_{V}$ and face points $p_{F}$. Tetrahedral sub-regions are defined by the span of a volume point and corresponding face points. Regularization ensures that the transformation is a diffeomorphism; that is, it is reversible and smooth. Hyper-elastic regularization, as defined by Burger et al. [20], is given by
(21)$$S^{hyper}\left(x\right)=\int \alpha_{1}\eta_{vol}\left(x\right)+\alpha_{2}\eta_{sur}\left(x\right)+\alpha_{3}\eta_{len}\left(x\right)d\Omega .$$
where $\alpha_{i}$ are balancing parameters. The functions $\eta_{vol}$, $\eta_{sur}$, and $\eta_{len}$ penalize changes of volume, surface, and length, respectively. Here, we set the balancing parameter as 10.0 for all experiments. Burger et al. [20] utilized the average points to delineate the volume point $p_{V}$ and six face points $p_{F}$. However, if the cage is concave (i.e., due to large deformations), the face and volume points are not maintained inside the cage, as shown in Figure 5. As a result, the functions $\eta_{vol}$ and $\eta_{sur}$ may have negative values, which can lead to the failure of gradient descent.

To achieve robust regularization, we define the face and volume vertices of each cage to have the same sub-area and sub-volume inside of the cage. Assuming that the face point $p_{F}=\left(p_{F_{x}},p_{F_{y}},p_{F_{z}}\right)$ is located inside the quadrilateral, the position $p_{F}$ of the points dividing the areas $▵p_{0}p_{1}p_{F}$, $▵p_{1}p_{2}p_{F}$, $▵p_{2}p_{3}p_{F}$, and $▵p_{3}p_{0}p_{F}$ is defined as follows;
(22)$$▵p_{i}p_{i+1}p_{F}=\left(p_{i+1}-p_{i}\right)\times \left(p_{F}-p_{i}\right)/2={\left[p_{i+1}-p_{i}\right]}_{\times }\left(p_{F}-p_{i}\right)/2,$$
where i={0,1,2,3}. The least-squares solution of the above conditions for all triangles is
(23)$$\begin{array}{c} \left[\begin{array}{c} {\left[p_{1}-p_{0}\right]}_{\times }\\ {}[p_{2}-p_{1}{]}_{\times }\\ {}[p_{3}-p_{2}{]}_{\times }\\ {}[p_{0}-p_{3}{]}_{\times } \end{array}\right]\left[\begin{array}{c} p_{F_{x}}\\ p_{F_{y}}\\ p_{F_{z}} \end{array}\right]=\left[\begin{array}{c} {\left[p_{1}\right]}_{\times }p_{0}\\ {}[p_{2}{]}_{\times }p_{1}\\ {}[p_{3}{]}_{\times }p_{2}\\ {}[p_{1}{0]}_{\times }p_{3} \end{array}\right]. \end{array}$$

Similar to the face point, we assume that the volume point is located inside the hexahedron. The volume point $p_{V}=\left(p_{V_{x}},p_{V_{y}},p_{V_{z}}\right)$ partitions 24 sub-tetrahedra of the cage. The volume of a single tetrahedron is given as
(24)$$V_{p_{i,j}p_{i+1,j}p_{F_{j}}}=\left(p_{i+1,j}-p_{i,j}\right)\times \left(p_{F_{j}}-p_{i,j}\right)\cdot \left(p_{V}-p_{i,j}\right)/6,$$
where $p_{F_{j}}$ is *j*th face point of the hexahedron and $p_{i,j}$ is the *i*th corner point of the *j*th face.

The volume point $p_{V}$ is obtained by solving the following least-squares problem:(25)$$\begin{array}{c} \left[\begin{array}{c} {\left[p_{1,0}-p_{0,0}\right]}_{\times }p_{F_{0}}-{\left[p_{1,0}\right]}_{\times }p_{0,0}\\ {}[p_{2,0}-p_{1,0}{]}_{\times }p_{F_{0}}-{\left[p_{2,0}\right]}_{\times }p_{1,0}\\ {}[p_{3,0}-p_{2,0}{]}_{\times }p_{F_{0}}-{\left[p_{3,0}\right]}_{\times }p_{2,0}\\ {}[p_{0,0}-p_{3,0}{]}_{\times }p_{F_{0}}-{\left[p_{0,0}\right]}_{\times }p_{3,0}\\ \vdots \\ {}[p_{0,5}-p_{3,5}{]}_{\times }p_{F_{5}}-{\left[p_{0,5}\right]}_{\times }p_{3,5} \end{array}\right]\left[\begin{array}{c} p_{V_{x}}\\ p_{V_{y}}\\ p_{V_{z}} \end{array}\right]=\left[\begin{array}{c} -{\left[p_{1,0}\right]}_{\times }p_{0,0}\cdot p_{F_{0}}\\ -{\left[p_{2,0}\right]}_{\times }p_{1,0}\cdot p_{F_{0}}\\ -{\left[p_{3,0}\right]}_{\times }p_{2,0}\cdot p_{F_{0}}\\ -{\left[p_{0,0}\right]}_{\times }p_{3,0}\cdot p_{F_{0}}\\ \vdots \\ -{\left[p_{0,5}\right]}_{\times }p_{3,5}\cdot p_{F_{5}} \end{array}\right]. \end{array}$$

The robust face/volume points improve the numerical stability of the cage deformation. The cost function is a combination of the dissimilarity measurement and regularization functions.

## 4. Interpolation of Shape Control Points

In this section, we introduce the shape interpolation and restoration for real-time usage of the 3D+t coronary artery model. According to Equation Equation 3, the shape of the coronary artery relies on the locations of the control points. Therefore, as we derive the intermediate positions of control points among cardiac phases, the corresponding shape is restored. To interpolate the positions of control points, we consider a set of control point at the $k^{\mathrm{th}}$ phase as a vector $P_{k}$, such that $P_{k}=\left\{p_{0},p_{1},\ldots ,p_{n-2},p_{n-1}\right\}$, where $p_{i}\in R^{3}$ and *n* is the number of cage control points. From the registration results, we interpolate the given sets of control points using periodic cubic spline interpolation [43], due to the (cyclic) nature of the heart’s motion. The number of knots is the same as the number of reconstructions from 4D CTA.

Let a phase-varying vector $S\left(t\right)=\{s_{0}\left(t\right),\ldots ,s_{n-1}\left(t\right)\}$ be the set of interpolated control points, where $s_{i}\left(t\right)$ is the *i*th control spline for the cardiac phase *t*. The spline vector S(t) has $C^{2}$ continuity with respect to the phase *t*. The spline function S(t) maps phase *t* to the set of cage control points, such that $S:R\to R^{3\times n}$. Then, the vertices of shape are restored using the following equation:(26)$$v\left(t\right)=F\left(v;S\left(t\right)\right)=\sum_{i\in I_{V}}\varphi_{i}\left(v\right)s_{i}\left(t\right).$$

## 5. Evaluations and Results

We evaluated the proposed method both qualitatively and quantitatively on data from eight patients. The proposed method was tested on an Intel (R) Xeon (R) W-2133 workstation with CPU@3.60 GHz and 32 GB ram. We partially multi-threaded the computation of cost function measurements using OpenMp [44] and Thread building block [45] during the optimization process. The proposed method and comparison target methods were written in C++.

### 5.1. Quantitative Evaluations

In the quantitative evaluation of non-rigid registration, we used metrics considering: (1) the closest point-mesh Euclidean distance (ED) from the target model to the matching result and (2) the dice coefficient (DC) obtained from the mesh boolean operation. As we set the number of iterations to 300/maxDepth for each depth, the total number of iterations for different max depths was set to be the same.

#### 5.1.1. Trade-off between Deformation Depth and Computation Time

First, we observed the trade-off between the degrees of freedom of deformation and computation time. As shown in Table 1, we compared the different depths of deformation incrementally, from 1 to 5. As shown in Figure 6, the ED and DC worsened, as the phases were far from the template phase. As the shape of the blood vessel was a thin tube shape, the ED and DC values noticeably deteriorated with slight movement. As the deformation depth increased, the ED values gradually decreased and the DC values increased more prominently; both metrics flattened for the other cardiac phases. The metrics converged after deformation depth 4. The comparison results for the other patients are given in Appendix A.

#### 5.1.2. Comparison with Other Methods

In the second experiment, the proposed registration method’s performance was compared with that of other non-rigid matching algorithms. As the comparison target of non-rigid registration, we selected the variations of GMM methods, which are combinations of a deformation model and a cost function. The deformation models were thin-plate spline (TPS) and generalized radial basis function (GRBF), while the cost functions were kernel correlation (KC) or $L_{2}$ distance. The comparison targets used L-BFGS-B as an optimization method.

For comparison, each deformation model had the same number of deformation control points. Considering the convergence of accuracy from the previous analysis, we set the number of grid and control points as 16×16×16 and 4913, respectively. As shown in Figure 7, the proposed method had higher DCs than the comparison targets at the interval [15%, 35%] of cardiac phases, where the interval had a DC value of less than 0.2 before registration. Although the ED metrics showed a similar trend, compared with other methods, a significant improvement in DC was observed. The comparison results for the other patients are given in Appendix A.

#### 5.1.3. Interpolation Accuracy

Furthermore, we evaluated the accuracy of the interpolated 3D+t coronary artery models by comparing them with the segmented models. The proposed method created a smooth and non-degenerate 3D model by interpolating the cage control points over sampled cardiac phases, as shown in Figure 8. Our data sets were evenly reconstructed from 4D CT within the R-R peak with 5% sampling interval. Thus, we had 20 keyframes ($V_{0\%}$, $V_{5\%}$, …, $V_{95\%}$). To evaluate the effect of sampling the cardiac phases, we chose phase sets from the given 20 keyframes as follows: (1) Odd 10: [5, 15, 25, …, 85, 95]; (2) Even 10: [0, 10, 20, …, 80, 90]; (3) Odd 5: [5, 25, 45, 65, 85]; and (4) Even 5: [0, 20, 40, 60, 80]. Figure 9 shows the differences among the phase selections. Although the sampled phase became sparse, the DC of the interpolated result showed that the results had a lower bound. The ED still showed a flattened value, when compared to that before registration. The comparison results for the other patients are given in Appendix A.

### 5.2. Qualitative Evaluations

In the qualitative evaluation, the role of the visualization effect of hyper-elastic regularization, geometrical comparison with other algorithms, comparison of the matching result and interpolation result model, and the limitations of the interpolation model were investigated.

#### 5.2.1. The Effect of Hyper-Elastic Regularization and Hierarchical Deformation

High-order deformation models often converge to a local minimum, which may look visually implausible. Figure 10a,b show examples of shape shrinkage when the target model contains loss of shape. The hyper-elastic regularization constraints lead to shape preservation, thus providing a plausible result, as shown in Figure 10c,d.

When compared with the other algorithms, Figure 11 shows an example where shape registration is defective at the excessively deformed and twisted parts. As the registration process converged to a local minimum, the deformation model represents the further details of local deformation. On the other hand, hierarchical cage deformation gradually acquired an optimal solution, passing from coarse to dense resolution, to avoid local minima, as shown in Figure 12. In this process, the low degree of freedom deformation serves as the initial value for the deformation in the next step. Therefore, local minima can be avoided more efficiently.

#### 5.2.2. The Representation Power of Interpolated Model

The proposed shape interpolation method may have limited representation ability for intermediate shapes. This limited shape representation is due to the recurring shape of the adjacent phases. We observed that the shape interpolation method restrictively delineates the intermediate shape. The shapes of the neighboring phases to the target phase resemble each other, but the target shape and the neighbor shapes are considerably different, as shown in Figure 13e.

## 6. Discussion and Conclusions

In this paper, we proposed a method for generating a 3D + time vessel model from 4D CT images that can be used in real-time. Our purpose was to create a 4D vascular model without mesh degeneration, interpolate the model at high speed, and express a more precise shape.

To create a 4D vascular model, we matched the diastolic coronary artery model with the coronary artery model in other phases through hierarchical cage deformation. During the registration process, hyper-elastic regularization was used as a shape preservation constraint. The shape control points obtained as a result of registration were interpolated into a cyclic cubic spline to create a 3D+t model. The shape change depends only on the control points of the cage. The rapid deformation application and the preservation features of the local information are beneficial in the shape registration process.

To evaluate the precision of the proposed method, quantitative and qualitative evaluation was performed on 160 CTA volumes acquired from eight patients. In the quantitative evaluation, we assessed:The trade-off between the shape matching accuracy and calculation time according to the hierarchical deformation;The comparative evaluation with other methods;The accuracy of the shape interpolation model, according to the time sampling interval.

In the step of measuring the shape matching precision according to the hierarchical deformation, we observed that the matching precision converged in the fourth step of the hierarchical deformation, where the calculation time was 23.53 s on average. In the fifth deformation depth of hierarchical transformation, the matching accuracy slightly increased, but the required time increased by 28.70% (to 33.00 s). In the hierarchical transformation of the cage creation stage, the control points constituting the cage increased exponentially, as a regular grid was used. We obtained the mean distance with precision of a 0.543 mm and standard deviation 0.222 in step 4 of the hierarchical transformation, where the Dice coefficient obtained an average of 0.754 and a standard deviation of 0.064.

Compared with other algorithms, the GMM method requires the creation of a mixture model for each point and, so, even with the same degree of freedom of transformation, the calculation time was as high as 40 s for the GRBF model and 33 s for the TPS model. In addition, in the average distance index, TPS_L2 was 0.530 mm, which had an error lower than that (0.543) of the proposed method; however, when comparing the Dice coefficient, TPS_L2 was observed to be 0.690, 0.045 points lower than the index of the proposed method (0.735).

If the indicators of DC and AD values conflict with each other, it is necessary to determine which indicator is better to express the accuracy of the matching result; for example, (1) high DC value and high AD value or (2) low DC Value and low AD value. At this time, we decided case (1) was a better indicator. This was because, in the vascular model between 15% and 35%, which showed a great difference with the 75% phase, a difference in AD values between the algorithms was not significantly observed, but the difference in DC values was noticeable. This was not only consistently observed in the quantitative evaluation of patient data, but also explains the differences arising from inappropriate deformations occurring in excessively twisted blood vessels during the qualitative evaluation.

The precision of the shape interpolation model was measured with different phase-sampling sets, where the accuracy was worsened with a larger phase-sampling interval. However, this limitation may be resolved by increasing the temporal resolution and by co-operating with the other real-time imaging systems, such as X-ray angiography or 4D US.

A qualitative evaluation was performed to support the quantitative evaluation mentioned above, as well as to observe the effect of the proposed model on the visualization stage. In the qualitative evaluation, (1) the effect of the modified hyper-elastic regularization and hierarchical transformation, and (2) the limitations in shape interpolation were observed.

When we observed the effect of hyper-elastic regularization, the shape was transformed to be visually plausible when hyper-elastic regularization was used. This phenomenon was particularly well seen in the coronary artery model with loss of shape. This is a characteristic obtained by minimizing the excessive deformation by robustly obtaining face points and volume points against the rapid degeneration of the cage mesh due to incorrect correspondence pairs. Compared with other non-rigid algorithms, the proposed method was able to cope with the local minima that occur during the optimization process by hierarchically performing the transformation effectively. In particular, it was shown that, in the vascular model with an excessive twist, optimal deformation was gradually obtained from low-dimensional deformation.

In summary, the proposed 3D+t vascular modeling method utilized hierarchical deformation for robust shape registration, while interpolation of the registered vascular structure enabled the restoration of small and complex geometries due to the cardiac cycle.

The electric potential of the myocardium generates the ECG signal, and the electric potential of the myocardium is related to movement and contraction of the heart. As an observation tool of the myocardium, the proposed method can provide an alternative to real-time imaging by using the ECG signal and 4D CT. In intraoperative situations, invasive coronary angiography is a method for monitoring the movement of the coronary artery, which provides limited deformation information (up to the 2D plane). With the proposed method, we expect that 3D contraction and strain of the myocardium, according to the ECG signal, can be observed, as the coronary arteries are attached to the epicardium. The ability of motion monitoring is directly related to the evaluation of the physiological function of the myocardium.

In addition, the modified hyper-elastic regularization prevented implausible deformation and mesh degeneration, which must be avoided in the analysis of 4D blood flow. We demonstrated the accuracy of the proposed method by presenting qualitative and quantitative evaluations using data from eight patients.

Therefore, we expect that the proposed 3D+t vascular model can be utilized in real-time applications such as (1) pre-operative blood flow analysis, which requires rapid shape creation without mesh degeneration; and (2) 2D invasive coronary angiography-3D shape registration during the intervention, where it can be used as an image guidance tool that provides real-time shape information according to the ECG signal during percutaneous coronary intervention.

As a limitation of this study, since the deformed model of the proposed model is limited to a 3D mesh, blood vessel segmentation is required in phases other than the template model. In a future study, we will conduct a study on a volume–template mesh model matching method which can be applied to volumetric data, including shape loss, in order to eliminate unnecessary repetitive processes.

## Figures and Tables

**Figure 1 sensors-20-05680-f001:**
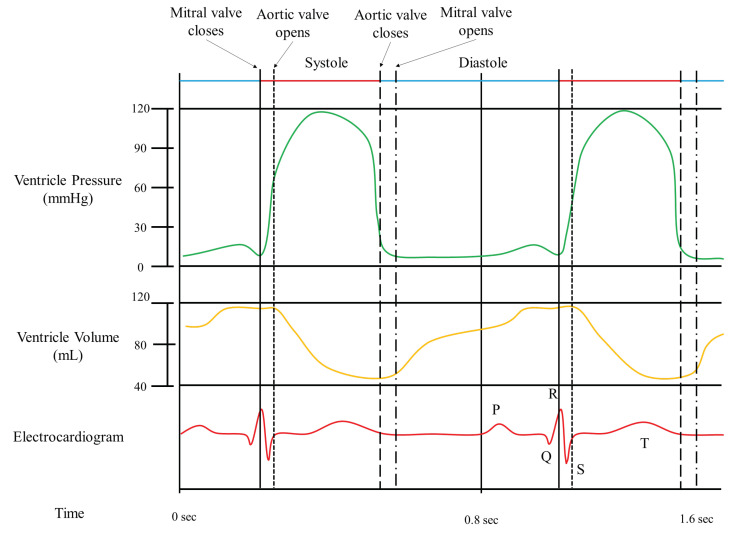
The ECG signal and volume of the ventricle during the different phases of a cardiac cycle.

**Figure 2 sensors-20-05680-f002:**
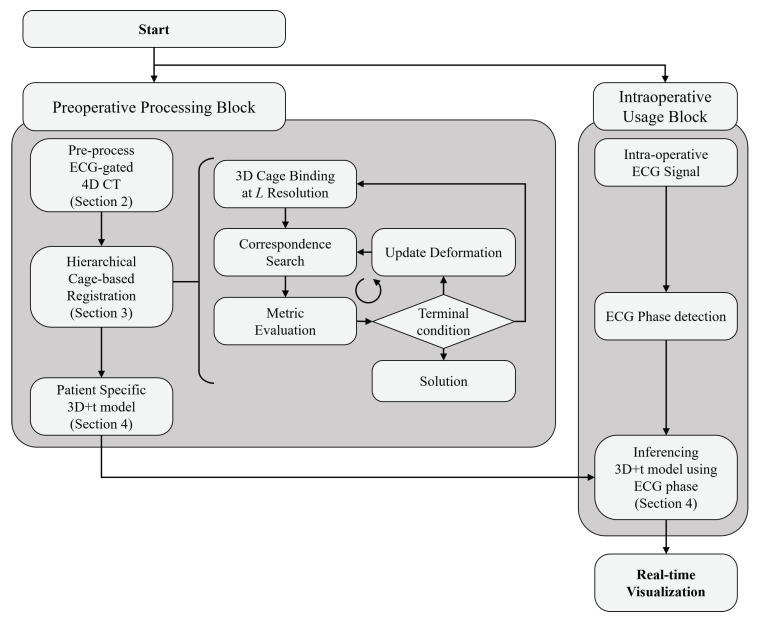
General framework of the proposed method.

**Figure 3 sensors-20-05680-f003:**
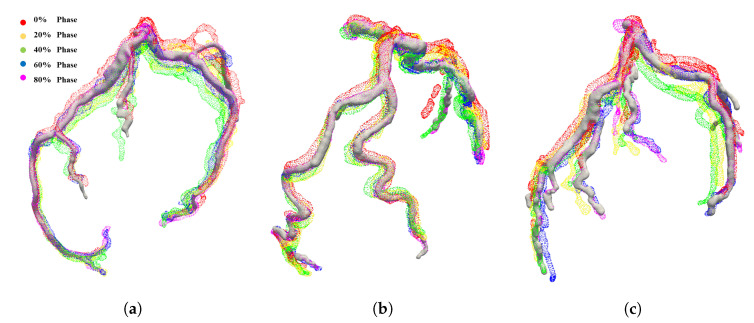
Left ascending and circumflex coronary arteries of: (**a**) patient 1; (**b**) patient 2; and (**c**) patient 4. The template phase model is a white solid model, while the other phases are colored with respect to their cardiac phases.

**Figure 4 sensors-20-05680-f004:**
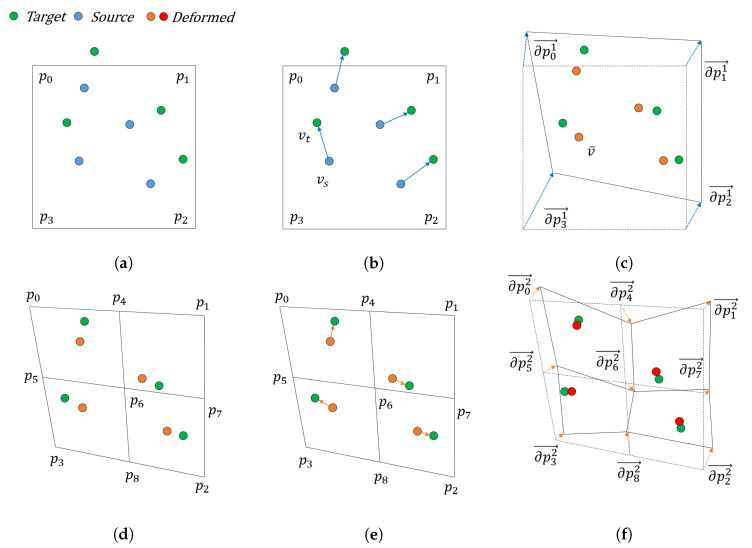
Hierarchical registration with different deformation depths. (**a**–**c**) Level 1; (**d**–**f**) Level 2 registration. (**a**,**d**) show cage partitioning at different levels. (**b**,**e**) show correspondence searching, while (**c**,**e**) show the gradient descent-based deformation update.

**Figure 5 sensors-20-05680-f005:**
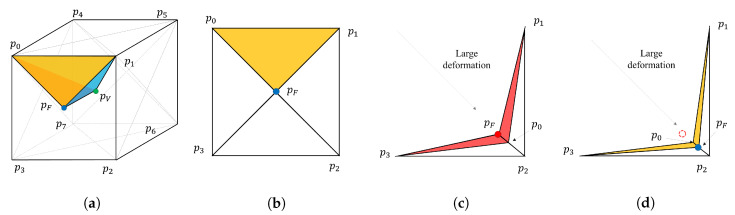
Cage sub-division with hyper-elastic regularization: (**a**) A tetrahedral sub-division of the 3D cage volume, which is the span of face (blue) and volume (green) points; (**b**) sub-division of cage face; (**c**) the average points (red) located outside of the cage and their negative areas (red triangles); and (**d**) the equal-area points (blue) located inside of cages despite large deformations and their positive areas (yellow triangles).

**Figure 6 sensors-20-05680-f006:**
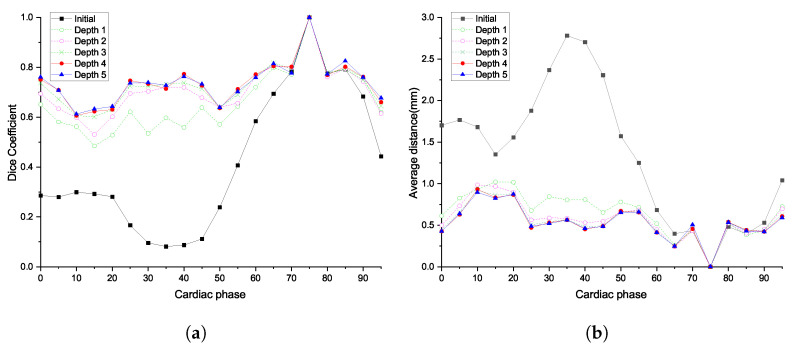
Effect of cage deformation depth for patient 1 in different cardiac phases: (**a**) dice coefficients; and (**b**) average distance from target to deformed model.

**Figure 7 sensors-20-05680-f007:**
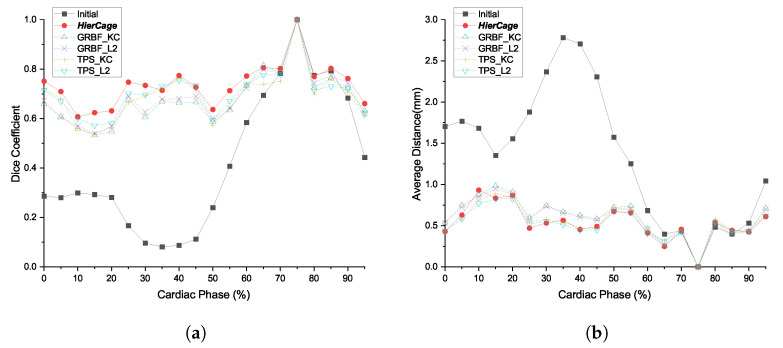
Comparison with other algorithms for patient 1 at the different cardiac phases: (**a**) dice coefficients; and (**b**) average distance from target to deformed model.

**Figure 8 sensors-20-05680-f008:**
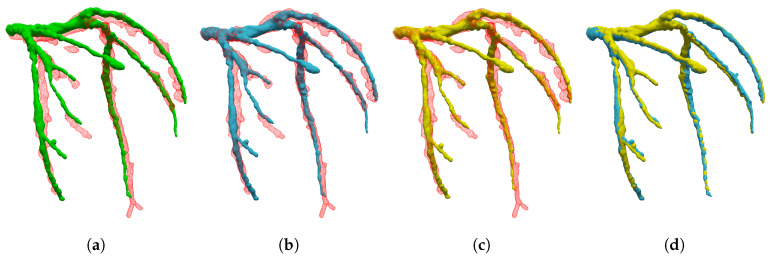
Comparison of the registered model and interpolated model for patient 5: (**a**) Template (green) and 40% coronary artery (red); (**b**) registered model (blue); (**c**) interpolated model (yellow); and (**d**) comparison of registered model and interpolated model.

**Figure 9 sensors-20-05680-f009:**
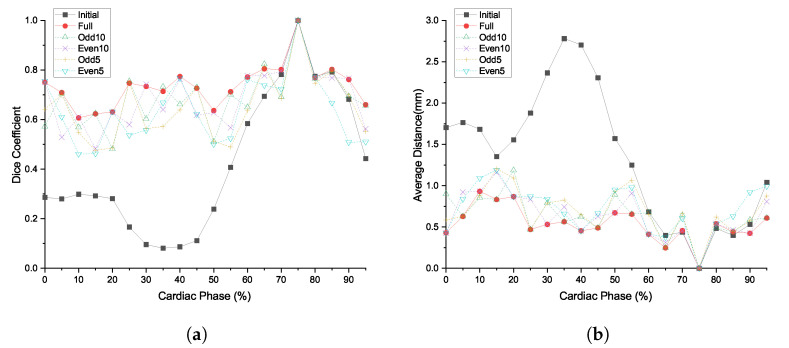
Comparison of interpolation sampling for patient 1 in the different cardiac phases: (**a**) dice coefficients; and (**b**) average distance from target to deformed model.

**Figure 10 sensors-20-05680-f010:**
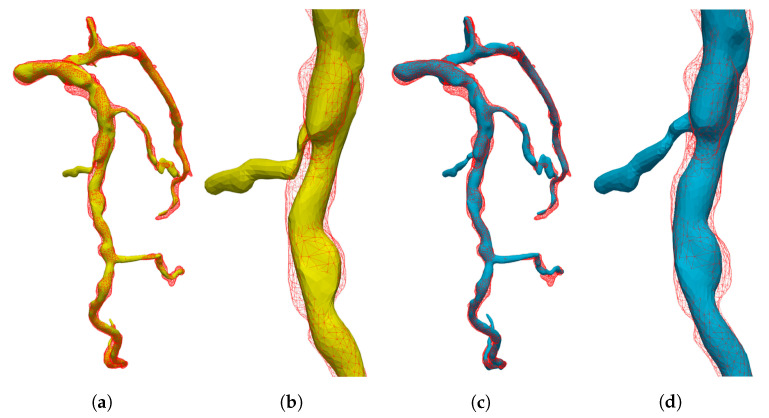
The effect of modified hyper-elastic regularization. We aligned the source model to the target model (red), which contains a loss of branch. The figures show effects: (**a**,**b**) without regularization (yellow) and (**c**,**d**) with regularization (blue).

**Figure 11 sensors-20-05680-f011:**
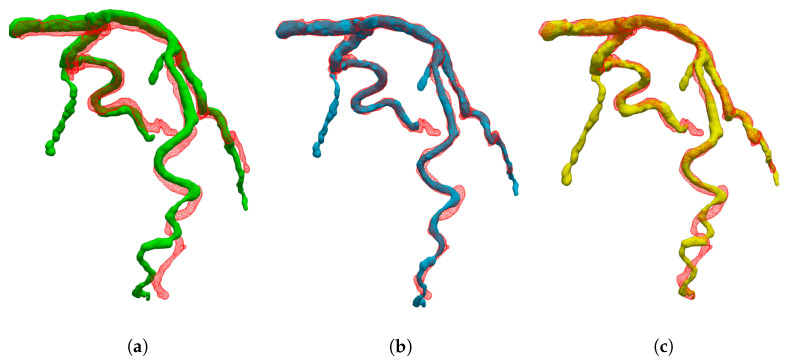
Qualitative comparison of GMM and the proposed method: (**a**) Initial template model (green) and target coronary artery (red); (**b**) result of the proposed method (blue); and (**c**) result of Gaussian mixture modeling with TPS+*L*_2_ (yellow).

**Figure 12 sensors-20-05680-f012:**
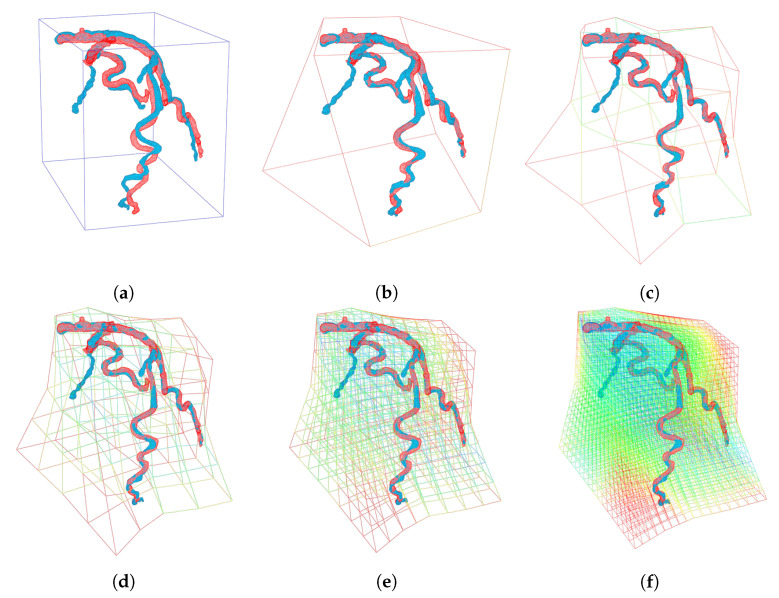
Qualitative comparison of the proposed method while changing the deformation resolution: (**a**) Initial template model (blue) and target model (red); (**b**–**f**) the results of registration (blue) at different cage resolutions from [1, 1, 1] to [5, 5, 5], respectively.

**Figure 13 sensors-20-05680-f013:**
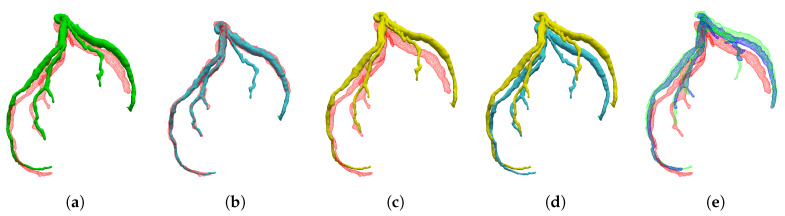
Comparison of the registered model and interpolated model for patient 8: (**a**) Template (green) and 95% coronary artery (red edges); (**b**) registered model (blue); (**c**) interpolated model (yellow); (**d**) comparison of registration interpolation; and (**e**) comparison of neighboring coronary artery models (90%, dark blue; 0%, light green).

**Table 1 sensors-20-05680-t001:** Trade-off between computation time and accuracy.

Method	Cage Resolution	Computation Time (s)	Average Distance (mm)	Dice Coefficient
HierCage	[1, 1, 1]	21.73	0.668 ± 0.255	0.655 ± 0.096
HierCage	[2, 2, 2]	23.05	0.597 ± 0.234	0.696 ± 0.077
HierCage	[3, 3, 3]	22.91	0.566 ± 0.227	0.721 ± 0.068
HierCage	***[4, 4, 4]***	***23.52***	***0.543 ± 0.222***	***0.735 ± 0.064***
HierCage	[5, 5, 5]	33.00	0.534 ± 0.221	0.741 ± 0.064
GRBF_KC	[4, 4, 4]	40.99	0.615 ± 0.218	0.666 ± 0.088
GRBF_L2	[4, 4, 4]	40.92	0.600 ± 0.207	0.671 ± 0.084
TPS_KC	[4, 4, 4]	33.00	0.553 ± 0.191	0.681 ± 0.080
TPS_L2	[4, 4, 4]	32.21	0.530 ± 0.17	0.690 ± 0.075

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
