# Peer review of "A Patient-Specific 3D+t Coronary Artery Motion Modeling Method Using Hierarchical Deformation with Electrocardiogram"

_sensors, 2020, doi:10.3390/s20195680_

Round 1

Reviewer 1 Report

The paperis very well written and the individual parts follow each other. The quality of the images/graphs is at a good level.
In chapter 5.1.2, lines 208-209, please improve the very general wording of the comparison results. On line 275, correct 0.45 to 0.045. Please specify what is better: to have a smaller average distance or a larger dice coefficient, and why?

Author Response

Dear Reviewer and MDPI-Sensors Editors,

Title: A patient-specific 3D+t coronary artery motion modeling method using a hierarchical deformation with electrocardiogram.

Journal: MDPI-Sensors

We would like to thank the reviewer for their interest and advice on the manuscript. Detailed comments helped improve the article. The manuscript was modified in consideration of the reviewer's recommendation. These changes were made mainly by:

- Background of the ECG phase and correlation with heart movement have been introduced

-Clarification, addition, and rewriting of the benefits of the proposed method.

- More practical utility of the proposed method has been introduced.

Thank you.

Comment 0.

The paper is very well written and the individual parts follow each other. The quality of the images/graphs is at a good level.

### Answer ###

Thank you for your kind attention to our article.

Comment 1.

In chapter 5.1.2, lines 208-209, please improve the very general wording of the comparison results.

### Answer ###

To help the reader's understanding, we rephrased the line 208-209 as follows;

from

"the proposed method shows equal or better accuracy and has faster than the other methods."

into lines 230-231,

"the proposed method had higher DCs than the comparison targets at the interval [15%, 35%]  of cardiac phases, where the interval had a DC value of less than 0.2 before registration.”.

Comment 2.

On line 275, correct 0.45 to 0.045.

### Answer ###

Because of our unintentional carelessness, the line 275 was a mistake. We correct the number 0.45 to 0.045 in the line 299.

Comment 3.

Please specify what is better: to have a smaller average distance or a larger dice coefficient, and why?

### Answer ###

In view of thoughtful advice, we extended the discussion and conclusion. Better indicators for expressing the accuracy of algorithms were discussed as follows:

In lines 300-307,

"If the indicators of DC and AD values conflict with each other, it is necessary to determine which indicator is better to express the accuracy of the matching result; for example, (1) high DC value and high AD value or (2) low DC Value and low AD value. At this time, we decided case (1) was a better indicator. This was because, in the vascular model between 15% and 35%, which showed a great difference with the 75% phase, a difference in AD values between the algorithms was not significantly observed, but the difference in DC values was noticeable. This was not only consistently observed in the quantitative evaluation of patient data, but also explains the differences arising from inappropriate deformations occurring in excessively twisted blood vessels during the qualitative evaluation."

Reviewer 2 Report

In this paper Authors propose creating a 3D+t cardiac coronary artery model that is rendered in real-time according to the ECG signal. Hierarchical cage-based deformation modeling is used to generate mesh deformation used during the procedure according to the ECG signal. It is nice paper, but in my opinion “Sensors” journal is not adequate journal.

I suggest to publish it in one of the following – more related to paper content journals:

  • Journal of Cardiovascular Development and Disease
  • Hearts
  • Healthcare
  • Diseases
  • Diagnostics
  • Medicines
  • Computation

Concerning the paper lot of cosmetics shortcomings (no spaces, units etc...) needs to be corrected – below few examples from 1-2 p. (but all pages needs to be investigated and corrected):

19 Cardiovascular disease(CVD)

43 the CTA, the CTA’s resolution is 50-200 ms which is lower than the 4D echocardiography (30-100hz)

44 and cardiac MRI(30-50 ms).

54 deformation. The most popular registration algorithm is the iterative closest point(ICP)

55 consists of finding the correspondence between twomodels and finding the optimal transformation[16]

Author Response

Dear Reviewer and MDPI-Sensors Editors,

Title: A patient-specific 3D+t coronary artery motion modeling method using a hierarchical deformation with electrocardiogram.

Journal: MDPI-Sensors

We would like to thank the reviewer for their interest and advice on the manuscript. Detailed comments helped improve the article. The manuscript was modified in consideration of the reviewer's recommendation. These changes were made mainly by:

- Background of the ECG phase and correlation with heart movement have been introduced

-Clarification, addition, and rewriting of the benefits of the proposed method.

- More practical utility of the proposed method has been introduced.

Thank you.

Comment 1.

In this paper Authors propose creating a 3D+t cardiac coronary artery model that is rendered in real-time according to the ECG signal. Hierarchical cage-based deformation modeling is used to generate mesh deformation used during the procedure according to the ECG signal.

It is nice paper, but in my opinion “Sensors” journal is not adequate journal.

### Answer ###

As kindly pointed out, we agree with the idea that the shape transformation of the coronary artery, one of the topics in this paper, may not be suitable for the Sensors.

But we saw that much of the proposed method fit within the scope of the 'Sensors'. This is because 3D + time dimension models are created and visualized according to real-time ECG signals acquired by processing continuous time-series information obtained from imaging techniques. To highlight the connection to the special issue of “Sensors for Vital Signs Monitoring,” a description of has been added.

The potential utility of the proposed method was additionally addressed as follows;

In lines 325-346

“In summary, the proposed 3D+t vascular modeling method utilized hierarchical deformation for robust shape registration, while interpolation of the registered vascular structure enabled the restoration of small and complex geometries due to the cardiac cycle.

The electric potential of the myocardium generates the ECG signal, and the electric potential of the myocardium is related to movement and contraction of the heart. As an observation tool of the myocardium, the proposed method can provide an alternative to real-time imaging by using the ECG signal and 4D CT. In intraoperative situations, invasive coronary angiography is a method for monitoring the movement of the coronary artery, which provides limited deformation information (up to the 2D plane). With the proposed method, we expect that 3D contraction and strain of the myocardium, according to the ECG signal, can be observed, as the coronary arteries are attached to the epicardium. The ability of motion monitoring is directly related to the evaluation of the physiological function of the myocardium.

In addition, the modified hyper-elastic regularization prevented implausible deformation and mesh degeneration, which must be avoided in the analysis of 4D blood flow. We demonstrated the accuracy of the proposed method by presenting  qualitative and quantitative evaluations using data from eight patients.

Therefore, we expect that the proposed 3D+t vascular model can be utilized in real-time applications such as (1) pre-operative blood flow analysis, which requires rapid shape creation without mesh degeneration; and (2) 2D invasive coronary angiography-3D shape registration during the intervention, where it can be used as an image guidance tool that provides real-time shape information according to the ECG signal during percutaneous coronary intervention.

Comment 2.

I suggest to publish it in one of the follows– more related to paper content journals:

Journal of Cardiovascular Development and Disease

Hearts

Healthcare

Diseases

Diagnostics

Medicines

Computation

### Answer ###

Since this paper was written from a technical point of view, the direct medical application of the proposed method is progressing as a future study. These studies may be much more suitable for the proposed journal.

Comment 3.

Concerning the paper lot of cosmetics shortcomings (no spaces, units etc...) needs to be corrected – below few examples from 1-2 p. (but all pages needs to be investigated and corrected):

19 Cardiovascular disease(CVD)

43 the CTA, the CTA’s resolution is 50-200 ms which is lower than the 4D echocardiography (30-100hz)

44 and cardiac MRI(30-50 ms).

54 deformation. The most popular registration algorithm is the iterative closest point(ICP)

55 consists of finding the correspondence between twomodels and finding the optimal transformation[16]

### Answer ###

Thank you for your careful reading and advice. Modifications, such as units and spaces, have been modified to follow recommendations. In addition, we have extensively modified English.

Reviewer 3 Report

This manuscript describes a methodology for aligning an interpolated 3D arterial model with ECG phase. In general, this paper has significant merit, but would be improved by addressing the following:

  1. Line 19: 22.2 million deaths from CVD starting when? Or is this meant that there will be 22.2 million deaths/year from CVD expected by 2030?
  2. What population is referred to on Line 69?
  3. Please define NHANES.
  4. I cannot decipher what Lines 25-26 are trying to communicate. Please rephrase.
  5. It would be valuable to have background on the ECG phases and how this might be reflected in the 3D vascular shape
  6. In general, it is unclear how this technology would be used or how it will provide a benefit. I think the authors are trying to communicate this, but additional commentary or rephrasing the existing text would help to clarify.
  7. In describing voxel resolution, please include units.
  8. Figure 1: there is some lack of clarity regarding where the process starts. As I look further, I see that we start with the 4D CT and the ECG signal, and this describes how those are processed
  9. Figure 2: I suggest adding a legend that correlates color with phase

Author Response

Dear Reviewer and MDPI-Sensors Editors,

Title: A patient-specific 3D+t coronary artery motion modeling method using a hierarchical deformation with electrocardiogram.

Journal: MDPI-Sensors

We would like to thank the reviewer for their interest and advice on the manuscript. Detailed comments helped improve the article. The manuscript was modified in consideration of the reviewer's recommendation. These changes were made mainly by:

- Background of the ECG phase and correlation with heart movement have been introduced

-Clarification, addition, and rewriting of the benefits of the proposed method.

- More practical utility of the proposed method has been introduced.

Thank you.

This manuscript describes a methodology for aligning an interpolated 3D arterial model with ECG phase. In general, this paper has significant merit, but would be improved by addressing the following:

Comment 1.

Line 19: 22.2 million deaths from CVD starting when? Or is this meant that there will be 22.2 million deaths/year from CVD expected by 2030?

###Answer###

As kindly pointed out, line 19 was changed to avoid confusion caused by incomplete information. The meaning of the sentence was clarified by adding the beginning year of statistics as follow;

“Cardiovascular disease (CVD) is one of the primary causes of death worldwide, with 22.2 million deaths expected from 2020 to 2030.”

Comment 2.

What population is referred to on Line 69?

### Answer ###

Here we used the word population as the training dataset. We rephrased to improve the reader's understanding as follow:

In line 75,

“On the other hand, the statistical shape deformation model (SSM) uses a low-dimensional statistical model, in which shape deformation is inferred from a training data set”

Comment 3.

Please define NHANES.

###Answer###

It was a mistake caused by our unintended carelessness, and we corrected what you kindly pointed out as follows:

In lines 19-20, “National Health and Nutrition Examination Survey (NHANES)”

Comment 4.

I cannot decipher what Lines 25-26 are trying to communicate. Please rephrase.

### Answer ###

As your advice, we rephrased the sentence to make it understandable as follows;

In lines 22-25,

"In particular, cardiovascular disease accounted for 14% of total medical spending in U.S., the highest rate among other major diagnostic groups-even higher than cancer. In the global population, the burden of expenditure is even more serious."

Comment 5.

It would be valuable to have background on the ECG phases and how this might be reflected in the 3D vascular shape

###Answer###

To allay concerns arising from missing the background to the ECG phase and its correlation with heart motion, a new figure and corresponding paragraphs were added to indicate the relationship between ECG signals and heart movements.

Figure 1 shows the correlation between ECG signals and heart movements. And paragraphs are given as follows:

In lines 26-37,

“Providing sufficient information through image analysis acquired in the pre-operative diagnosis stage eliminates unnecessary examination and helps in developing patient-specific treatment plans. As the heart is a continuously beating organ and there may be unexpected movements in patients (e.g., arrhythmia), if information on this movement can be obtained in advance, coronary artery and heart procedures may be more efficient.

As shown in Figure 1, the ECG signal is a change in potential that is correlated with the movement of the heart muscle. Motion of the heart produces changes in volume and pressure in the cardiac chambers; therefore, ECG provides important information about the movement of the heart. When CT is reconstructed by performing retrospective ECG synchronization, the movements of the heart and coronary arteries (according to the cardiac cycle) can be obtained geometrically, and movement information (e.g., coronary artery distortion), as well as characteristics of the coronary stenosis, can be obtained.”

Comment 6.

In general, it is unclear how this technology would be used or how it will provide a benefit. I think the authors are trying to communicate this, but additional commentary or rephrasing the existing text would help to clarify.

###Answer###

To elaborate on the use of the proposed method and its advantages, discussion sections were expanded by explicitly mentioning uses as follows:

In lines 325-346

“In summary, the proposed 3D+t vascular modeling method utilized hierarchical deformation for robust shape registration, while interpolation of the registered vascular structure enabled the restoration of small and complex geometries due to the cardiac cycle.

The electric potential of the myocardium generates the ECG signal, and the electric potential of the myocardium is related to movement and contraction of the heart. As an observation tool of the myocardium, the proposed method can provide an alternative to real-time imaging by using the ECG signal and 4D CT. In intraoperative situations, invasive coronary angiography is a method for monitoring the movement of the coronary artery, which provides limited deformation information (up to the 2D plane). With the proposed method, we expect that 3D contraction and strain of the myocardium, according to the ECG signal, can be observed, as the coronary arteries are attached to the epicardium. The ability of motion monitoring is directly related to the evaluation of the physiological function of the myocardium.

In addition, the modified hyper-elastic regularization prevented implausible deformation and mesh degeneration, which must be avoided in the analysis of 4D blood flow. We demonstrated the accuracy of the proposed method by presenting qualitative and quantitative evaluations using data from eight patients.

Therefore, we expect that the proposed 3D+t vascular model can be utilized in real-time applications such as (1) pre-operative blood flow analysis, which requires rapid shape creation without mesh degeneration; and (2) 2D invasive coronary angiography-3D shape registration during the intervention, where it can be used as an image guidance tool that provides real-time shape information according to the ECG signal during percutaneous coronary intervention.”

Comment 7.

In describing voxel resolution, please include units.

### Answer ###

Modification of units and spaces, including voxel resolution, has been modified according to advice. It also modified the double spacing of words and unclear capital letters.

Comment 8.

Figure 1: there is some lack of clarity regarding where the process starts. As I look further, I see that we start with the 4D CT and the ECG signal, and this describes how those are processed

### Answer ###

To provide the reader's better understanding, we have redrawn the figure to including start and end. In addition, we explicitly noted the pre-operative processes and intra-operative processes. The details of each procedure have been described as following;

In lines 109-127,

“In this paper, we propose a 3D+t coronary artery model that can be inferred in real-time, according to ECG signals. The overall structure of the proposed method is shown in Figure 1. The proposed patient-specific 3D+t coronary artery motion model is divided into two processing blocks, according to the timing of data processing; (1) a preoperative processing block, and (2) an intraoperative usage block.

At the preprocessing step, we first perform the segmentation of 4D CTA volumes to generate artery models. After we have multiple coronary artery models, hierarchical cage-based registration is performed to construct a patient-specific 3D+t model with hyperelastic regularization. The proposed hierarchical cage deformation model more robustly/accurately registers coronary artery models in different cardiac phases. When updating the control points of the coronary artery model, we gradually increase the degrees of freedom of the deformation model. A modified hyper-elastic regularization term prevents mesh degeneration problems during the control point optimization step. After the optimal cage control point is obtained---which minimizes the shape dissimilarity of the source shape and the target shape---we interpolate the shape control point to build a continuous 3D+t model. The interpolated shape model provides fast shape-inference for intraoperative usage.

In the intraoperative usage step, the ECG phase can be assessed from the patient's real-time signal, which is then correlated to the geometric deformation of cardiac muscles, as shown in Figure 2. Therefore, the 3D shape of the coronary artery is provided by a continuous 3D+t model, according to change of the ECG signal and time.”

Comment 9.

Figure 2: I suggest adding a legend that correlates color with phase

### Answer ###

We added a color label for each heart phase, with additional examples presented in the picture.
